# Are People More Averse of Their Peers Living in Hardship or Driving Luxury Cars? Individuals’ Willingness to Accept Their Peers’ Relative Circumstances

**DOI:** 10.3390/bs15101395

**Published:** 2025-10-15

**Authors:** Xiaohan Hu, Jie Liu, Fengyang Sun, Xiuxin Wang

**Affiliations:** School of Psychology, Qufu Normal University, Qufu 273165, China; xhhu0412@foxmail.com (X.H.);

**Keywords:** social comparison, Tri-Reference Point theory, minimum requirement, goal, relative deprivation

## Abstract

Are people more afraid of peers living in hardship or peers driving luxury cars? Tri-Reference Point theory posits that individuals prioritize others’ minimum requirements over their goals. This suggests people should be less willing to accept peers’ minimum requirements are not being met (i.e., living in hardship) than peers achieving goals (i.e., driving luxury cars). However, four experiments (*N* = 648) revealed that in social comparison contexts, people exhibit greater reluctance toward peers “driving luxury cars” (Experiments 1–4). This phenomenon occurs because peers “driving luxury cars” triggers stronger relative deprivation in individuals (Experiments 1–2). When situational competitiveness diminishes or demonstrating peers’ effort, willingness to accept peers “driving luxury cars” increases (Experiments 3–4). Theoretically, these findings indicate that under social comparison, the psychological weighting of others’ goals versus minimum requirements reverses—individuals become more concerned with whether others achieve goals than whether they meet minimum requirements. Practically, this study offers insights for enhancing the acceptance of others’ high achievement and promoting team harmony.

## 1. Introduction

The Chinese saying, “We’re not only afraid of our peers’ living in hardship, but also afraid of them driving luxury cars,” reveals complex attitudes toward others’ situations. On one hand, witnessing peers suffer can evoke sympathy, driven by concern for their well-being. On the other hand, observing peers achieve success while one remains in a mediocre state can trigger feelings of envy ([5]; [16]). Psychological research also provides supporting evidence for this: on the one hand, driven by general altruism, most people are willing to help peers, or even strangers, in distress, regardless of whether they receive anything in return ([12]; [14]); on the other hand, people also envy their peers’ successes and may even engage in anti-social behaviors as a result ([43]; [6]). However, do individuals fear their peers’ “living in hardship” or “driving luxury cars” more?

### 1.1. Tri-Reference Point Theory Under Social Comparison

Whether individuals fear their peers’ “living in hardship” or “driving luxury cars” more essentially reflects their acceptance willingness towards their peers’ situations. Research in judgment and decision-making indicates that when making comparisons and evaluations, people select specific values or standards as reference points ([25]; [47]). The Tri-Reference Point (TRP) theory further proposes that individuals generally treat the minimum requirement, status quo and goal as reference points. Moreover, people assign different psychological weights to these three reference points; typically, avoiding falling below the minimum requirement is most important, followed by achieving a goal, and finally by making changes relative to the status quo ([38]; [39]). Research also provides supporting evidence for the distinct psychological weights people assign to the minimum requirement and goal reference points. For instance, when considering situations concerning themselves, people may forgo risky options that could potentially help them achieve a goal to avoid falling below the minimum requirement ([38]). When considering others’ situations, individuals are more willing to help others avoid falling below the minimum requirement than to help them achieve a goal ([20]). According to TRP theory, peers’ “living in hardship” typically represents situations where individuals fall below the minimum requirement—such as experiencing investment losses, being fired from a company, or ranking at the bottom in exams—whereas peers “driving luxury cars” generally reflects goal-attainment scenarios, such as achieving substantial investment returns, receiving consecutive promotions, or ranking at the top in academic assessments. Since the psychological weight of the minimum requirement is greater than that of the goal, people should be less willing to accept peers’ “living in hardship”.

However, in the problem context of fearing peers’ “living in hardship” or “driving luxury cars”, individuals not only care about their peers’ situations but also consider the comparison between their own situation and that of their peers. In other words, this dilemma contains a critical implicit assumption: individuals implicitly use their current status as the status quo reference point. A peer “living in hardship” implies the peer is worse off than oneself, while a peer “driving luxury cars” implies the peer is better off.

Social comparison theory posits that comparison is ubiquitous, and people experience different feelings based on the outcomes of comparing themselves to others ([16]; [27]). For instance, when an individual’s investment return is at a relative disadvantage in comparison, even if they are profitable overall, they still perceive the situation as a loss ([19]). This suggests that people consider not only others’ absolute situations but also evaluate the relative situations of themselves and others through the lens of social comparison. Yet, previous research on the TRP theory primarily focused on how people view their own or others’ absolute situations. When social comparison information is introduced, do people still assign the same psychological weights to others’ “minimum requirement” and “goal” reference points? Do people still prioritize others’ minimum requirements, meaning they remain more averse to seeing others “living in hardship”?

### 1.2. Peers’ Success Induces Stronger Relative Deprivation than Peers’ Suffering

Comparing with peers’ “living in hardship” or “driving luxury cars” can be conceptualized as downward versus upward social comparisons. During downward comparisons, individuals predominantly focus on personal gains rather than situational inequalities ([8]; [42]). In upward comparisons, however, people emphasize self–other discrepancies and perceive relative losses ([26]; [34]).

Psychological experiences arising from social comparison are independent of individuals’ absolute gains or losses, but are instead shaped by others’ relative gains. When others’ gains exceed one’s own, individuals perceive greater losses and experience relative deprivation (see [37]; [46]). This effect persists even when comparators lack similarity to the individual, as the mere disparity in outcomes remains sufficient to trigger such psychological responses ([26]). The feeling of relative deprivation is related to the fact that people feel they deserve the object they did not obtain and that they find this unfair (i.e., they are competent and deserve this gain). For instance, when a peer achieves success, an individual may feel discontent regarding their own lack of comparable success, thereby experiencing relative deprivation. Empirical evidence further indicates that upward social comparisons trigger stronger relative deprivation than downward comparisons ([29]; [30]). Thus, in social comparison contexts, peers’ goal-attaining “driving luxury cars” may amplify individuals’ relative deprivation, rendering them less willing to accept such situations. Based on this, we propose the following hypotheses:

**Hypothesis** **1.**
*Compared with peers’ “living in hardship” (falling below the minimum requirements), individuals will show lower acceptance willingness towards peers’ “driving luxury cars” (achieving the goals).*


**Hypothesis** **2.**
*Relative deprivation mediates the relationship between peers’ circumstances and individuals’ acceptance willingness towards those situations. Specifically, compared with peers’ “living in hardship”, peers’ “driving luxury cars” will trigger a higher level of relative deprivation, resulting in greater unwillingness to see this situation occur.*


### 1.3. Moderating Effects of Situational Competitiveness and Peers’ Effort Level

When individuals find themselves at a disadvantage in social comparison, their altruistic tendency diminishes, thereby impairing collective interests ([28]; [49]). Therefore, to maximize overall welfare, it is necessary to enhance people’s acceptance willingness toward peers’ “driving luxury cars”. The question then arises: under what conditions are individuals more inclined to accept peers’ “driving luxury cars”?

First, the competitiveness between comparison targets is a salient feature of social comparison ([21]). In highly competitive situations, individuals exhibit a stronger tendency to compare themselves with others ([33]). Conversely, when the situational competitiveness is low, individuals exhibit reduced motivation to compare themselves with others ([7]) and are less likely to engage in harmful behaviors driven by envy ([6]). Research also indicates that competitiveness can lead people to perceive others’ successes as being achieved at their own expense, thereby triggering a stronger sense of relative deprivation ([29]; [35]). Therefore, under less competitive situations, the relative deprivation elicited by peers’ success is weaker, leading to greater acceptance of peers who are “driving luxury cars”. Based on this, we hypothesize the following:

**Hypothesis** **3.**
*The competitiveness of a situation moderates individuals’ acceptance willingness toward peers’ “driving luxury cars”. When competitiveness is low, peers’ success triggers lower levels of relative deprivation, thereby increasing acceptance willingness. Conversely, in highly competitive contexts, peers’ “driving luxury cars” elicits stronger relative deprivation, consequently reducing acceptance willingness.*


Second, during social comparisons, individuals consider not only the comparison of outcomes but also the comparison of inputs between themselves and others ([1]). Individuals tend to imbue effort with moral value, perceiving those who exert effort as possessing superior moral qualities ([9]). Consequently, they exhibit greater acceptance of success and resulting inequalities when such outcomes are perceived as earned through effort ([11]; [15]). This suggests that if peers’ “driving luxury cars” is attributed to high effort, the associated relative deprivation would diminish, thereby enhancing acceptance of peers’ success. Based on this, we hypothesize the following:

**Hypothesis** **4.**
*Peers’ effort level moderates individuals’ acceptance willingness of peers’ “driving luxury cars”. When success is attributed to high effort, it evokes weaker relative deprivation, leading to higher acceptance willingness. Under control conditions, however, peers’ success provokes stronger relative deprivation and lower acceptance willingness.*


### 1.4. Overview of Studies

Through four experiments, we examined whether individuals exhibit lower acceptance willingness toward peers’ “driving luxury cars” compared to peers’ “living in hardship”, along with the mediating and moderating mechanisms underlying this effect. Experiment 1 first tested this phenomenon using a lottery scenario with a university student sample, investigating both the difference in acceptance willingness and the mediating role of relative deprivation. Subsequently, Experiment 2 recruited a community-based sample to validate these findings in a bonus allocation scenario, thereby enhancing ecological validity. Experiment 3 and 4 then explored the moderating roles of situational competitiveness and peers’ effort level, respectively, aiming to provide actionable insights for increasing acceptance willingness toward peers’ “driving luxury cars”.

## 2. Experiment 1: Buy Lottery Tickets

Experiment 1 recruited undergraduate participants through a university research pool in China to preliminarily test whether individuals exhibit lower acceptance willingness of peers’ “driving luxury cars” compared to “living in hardship”, while examining the mediating role of relative deprivation.

### 2.1. Pilot Experiment

Before conducting the formal experiment, we first operationalized the definitions of “living in hardship” and “driving luxury cars” for participants. “Living in hardship” is typically associated with losses, whereas “driving luxury cars” is associated with gains. Therefore, we asked participants to specify what magnitude of gains or losses would be perceived as constituting “living in hardship” or “driving luxury cars”.

The pilot experiment recruited 29 participants (21 females; *M*_age_ = 20.24, *SD* = 1.66). Participants imagined that both they and a friend spent 1000 CNY on lottery tickets, with themselves breaking even (total gain = 0 CNY). They then indicated the loss/gain amounts that would qualify the friend’s outcome as “living in hardship” or “driving luxury cars”.

Using median responses, results showed that for a 1000 CNY lottery investment: A peer’s loss of 500 CNY was perceived as “living in hardship”; a peer’s gain of 10,000 CNY was perceived as “driving luxury cars”. These values were adopted as experimental manipulations.

### 2.2. Participants and Design

According to G*Power 3.1 calculations, for an independent-samples *t*-test with a medium effect size (Cohen’s *d* = 0.5), one-tailed testing, and 80% statistical power, a minimum sample size of 102 participants was required. A total of 140 participants were recruited (80 females; *M*_age_ = 23.61, *SD* = 4.94). A single-factor between-subjects design was adopted, with the independent variable being peers’ relative circumstances (“living in hardship” vs. “driving luxury cars”) and the dependent variable being participants’ acceptance willingness of peers’ relative circumstances.

### 2.3. Procedure

First, participants were randomly assigned to two conditions and completed demographic questions. They were asked to recall a close friend and an activity they did together, and then write down the friend’s name.

Next, participants imagined the following: you and this friend each spent 1000 CNY on lottery tickets. You won 1000 CNY (breaking even). Depending on the assigned condition, participants read either: “Your friend won 500 CNY (a loss of 500 CNY)” or “Your friend won 11,000 CNY (a gain of 10,000 CNY).”

Subsequently, relative deprivation was measured across cognitive and affective dimensions. Participants rated on 7-point scales: “Compared to your friend’s outcome, how would you evaluate your own outcome?” (1 = very unfavorable, 7 = very favorable) and “How satisfied are you with your outcome compared to your friend’s?” (1 = very dissatisfied, 7 = very satisfied) ([48]). They also rated “To what extent are you willing to see such a situation occur?” (1 = very unwilling, 7 = very willing). Additionally, social desirability bias could influence participants’ choices, leading them to appear more averse to peers living in hardship and more accepting of peers driving luxury cars. Consequently, after finishing all tasks, participants completed a short-form social desirability scale to be included as a covariate ([41]).

### 2.4. Results and Discussion

An independent-samples *t*-test with acceptance willingness of peers’ relative circumstances as the dependent variable revealed significantly lower willingness of peers’ “driving luxury cars” (*M* = 2.77, *SD* = 1.57) compared to “living in hardship” (*M* = 3.40, *SD* = 1.87), *t* (138) = 2.15, *p* = 0.03, Cohen’s *d* = 0.36. When controlling for social desirability using ANCOVA, the effect remained significant, *F* (1, 137) = 4.62, *p* = 0.03, η^2^_p_ = 0.03.

The relative deprivation items demonstrated high internal consistency (α = 0.92). Reverse-scored items were averaged for analysis, with higher scores indicating stronger relative deprivation. An independent-samples *t*-test revealed significantly stronger relative deprivation in the “driving luxury cars” condition (*M* = 4.27, *SD* = 1.46) compared to the “living in hardship” condition (*M* = 1.60, *SD* = 1.30), *t* (138) = 11.42, *p* < 0.001, Cohen’s *d* = 1.93.

A bootstrap mediation analysis using the PROCESS ([23]) with 5000 resamples and 95% confidence intervals revealed a significant indirect effect of relative deprivation (95% *CI* = [−1.47, −0.26]). When peers were “driving luxury cars”, it triggered stronger relative deprivation, thereby reducing acceptance willingness (Figure 1).

Experiment 1 preliminarily validated Hypotheses 1 and 2: compared to peers’ “living in hardship”, individuals exhibited lower acceptance willingness of peers’ “driving luxury cars”, which was driven by the heightened relative deprivation evoked by peers’ “driving luxury cars”.

## 3. Experiment 2: Bonus Allocation

Experiment 1 revealed that compared with peers’ “living in hardship”, people were less willing to accept peers’ “driving luxury cars”. However, as participants in Experiment 1 were primarily university students, the findings may have limited generalizability. Therefore, starting from Experiment 2, the participant recruitment was not restricted to university students; instead, we utilized the Credamo platform sample pool for online participant recruitment. Furthermore, Experiment 2 employed a different scenario to enhance the ecological validity of the research.

### 3.1. Participants and Design

Experiment 2 replicated the research design and sample size calculation protocol from Experiment 1. A total of 120 participants were recruited (83 females; *M*_age_ = 31.03, *SD* = 7.80).

### 3.2. Procedure

Participants were randomly assigned to two conditions. After completing demographic questions, they read the following: You and a same-sex friend are employees at the same company. Both of you worked equally hard for a year. After the year-end bonus announcement, you received 1000 CNY, while your friend received 500 CNY (“living in hardship”)/ 10,000 CNY (“driving luxury cars”).

Finally, participants completed the identical measures of relative deprivation and acceptance willingness as in Experiment 1.

### 3.3. Results and Discussion

An independent-samples *t*-test with acceptance willingness as the dependent variable revealed a significant effect of peers’ relative circumstances. Participants showed lower willingness of peers’ “driving luxury cars” (*M* = 2.33, *SD* = 1.45) compared to “living in hardship” (*M* = 4.45, *SD* = 1.67), *t* (118) = 7.42, *p* < 0.001, Cohen’s *d* = 1.35.

A *t*-test with relative deprivation as the dependent variable revealed significantly higher relative deprivation in the “driving luxury cars” condition (*M* = 5.73, *SD* = 1.22) compared to the “living in hardship” condition (*M* = 2.13, *SD* = 0.94), *t* (118) = 18.08, *p* < 0.001, Cohen’s *d* = 3.30.

With the peers’ relative circumstances as the independent variable and participants’ acceptance willingness toward the situation as the dependent variable, bootstrap mediation analysis using the PROCESS confirmed the significant mediating role of relative deprivation (Figure 2), 95% *CI* = [−2.92, −1.25].

Experiment 2 further validated the conclusion of Experiment 1: compared to peers’ “living in hardship” (falling below the minimum requirements), individuals exhibited significantly lower acceptance willingness toward peers’ “driving luxury cars” (achieving the goals). This effect is driven by stronger relative deprivation triggered by peers’ “driving luxury cars”.

## 4. Experiment 3: The Moderating Effect of Situational Competitiveness

Study 3 manipulated situational competitiveness to examine whether this situational factor enhances individuals’ acceptance willingness of peers’ “driving luxury cars”.

### 4.1. Participants and Design

According to G*Power calculations, for a two-way ANOVA with a medium effect size (f = 0.25), one-tailed testing and 80% statistical power, a minimum sample size of 179 participants was required. We recruited 208 participants (156 females; *M*_age_ = 31.14, *SD* = 8.00). The experiment employed a 2 (peers’ relative circumstances: “living in hardship” vs. “driving luxury cars”) × 2 (situational competitiveness: high vs. low) between-subjects factorial design, with acceptance willingness as the dependent variable.

### 4.2. Procedure

Participants were randomly assigned to conditions and completed demographic questions. Those in high/low competitiveness conditions read the following: You and your friend are employees at a company. A bonus pool will be distributed. The amounts are determined by employee competition (you both want more)/by job rank and seniority (your position is lower/higher and tenure shorter/longer than your friend’s).

Subsequently, all participants read the following: You received 1000 CNY, while your friend received 500 CNY (“living in hardship”)/ 10,000 CNY (“driving luxury cars”).

Finally, consistent with prior findings that situational competitiveness promotes zero-sum beliefs ([29]), we included a zero-sum belief item (‘When some people are getting poorer, it means that other people are getting richer.’) on a 1–7 point scale ([32]) to serve as a manipulation check, since participants’ inherent altruism could influence the results. For example, highly altruistic individuals are more willing to help those in hardship and more accepting of others’ success ([5]). We also measured altruistic tendencies using a dictator game task ([18]). Specifically, participants imagined completing a task with a stranger with equal effort and time invested. They were informed that a total reward of 10 CNY was to be divided between themselves and the stranger, and they indicated how much they would allocate to the other party.

### 4.3. Results and Discussion

The manipulation check revealed significantly higher zero-sum belief scores in the high-competitiveness group (*M* = 4.81, *SD* = 1.68) versus the low-competitiveness group (*M* = 3.67, *SD* = 1.86), *t* (206) = 4.63, *p* < 0.001, Cohen’s *d* = 0.64, confirming the effectiveness of the competitiveness manipulation.

A two-way ANOVA was performed with peers’ relative circumstances and situational competitiveness as independent variables and acceptance willingness toward the situation as the dependent variable. The interaction between peers’ relative circumstances and situational competitiveness was significant, *F* (1, 204) = 4.50, *p* = 0.04, η^2^_p_ = 0.02. Simple effect analysis (Figure 3) indicated that under the “driving luxury cars” condition, participants exhibited significantly lower acceptance willingness in high competitiveness (*M* = 1.81, *SD* = 0.17) compared to low competitiveness (*M* = 2.61, *SD* = 0.18), *F* (1, 204) = 10.70, *p* = 0.001, η^2^_p_ = 0.05. In contrast, under the “living in hardship” condition, no significant difference emerged between high (*M* = 4.74, *SD* = 0.17) and low competitiveness groups (*M* = 4.79, *SD* = 0.19), *F* (1, 204) = 0.04, *p* = 0.84. After controlling for dictator game scores as a covariate, all main effects and interactions remained significant (*ps* < 0.05).

A moderated mediation analysis was conducted using the PROCESS (Model 7), with peers’ relative circumstances as the independent variable, acceptance willingness as the dependent variable, relative deprivation as the mediator, and situational competitiveness as the moderator. The results established a significant moderated mediation model (Table 1), β = 1.78, *t* (204) = −2.12, *p* < 0.001, 95% *CI =* [1.17, 2.4]. Under low-competitiveness conditions, peers’ relative circumstances exerted a significant mediating effect on acceptance willingness through relative deprivation, 95% *CI* = [−1.77, −0.99]; under high-competitiveness conditions, this mediating effect remained significant, 95% *CI* = [−2.91, −1.85]. Critically, however, the mediating effect of relative deprivation was significantly stronger in high-competitiveness conditions compared to low-competitiveness conditions, 95% *CI* = [−1.49, −0.61]. This indicates that when situational competitiveness is low, peers “driving luxury cars” elicit lower levels of relative deprivation, thereby enhancing individuals’ acceptance of such situations.

Experiment 3 supported Hypothesis 3, demonstrating that individuals show higher acceptance willingness of peers’ “driving luxury cars” in low-competitiveness contexts. Crucially, the acceptance willingness of peers’ “living in hardship” remained unaffected by competitiveness levels.

## 5. Experiment 4: The Moderating Effect of Peers’ Effort Levels

This experiment manipulated peers’ effort levels to examine its moderating effect on individuals’ acceptance willingness of peers’ “driving luxury cars”.

### 5.1. Participants and Design

Experiment 4 employed the same research design and sample size calculation procedure as Experiment 3. We recruited 180 participants (126 females; *M*_age_ = 29.43, *SD* = 7.72). The study used a 2 (peers’ relative circumstances: “living in hardship” vs. “driving luxury cars”) × 2 (peers’ effort level: high effort vs. control) between-subjects design, with acceptance willingness as the dependent variable.

### 5.2. Procedure

Participants were randomly assigned to conditions and completed demographic measures. All read as follows: You and your friend are employees at the same company. Those in the high-effort condition were informed that “throughout the year, he/she devoted more effort than you to work,” while control condition participants received no such information.

Subsequently, participants read the following: After the year-end bonus announcement, you received 1000 CNY, while your friend received 500 CNY (“living in hardship”) or 10,000 CNY (“driving luxury cars”).

Then, participants conducted measures of relative deprivation and acceptance willingness toward the situation identical to those in previous experiments. Finally, participants completed the same measures of relative deprivation and situational acceptance willingness as in previous studies, along with the same altruism measurement (i.e., the dictator game) used in Experiment 3.

### 5.3. Results and Discussion

A two-way ANOVA was performed with peers’ relative circumstances and effort level as independent variables and acceptance willingness toward the situation as the dependent variable. The interaction between peers’ relative circumstances and effort level was significant, *F* (1, 176) = 5.25, *p* = 0.02, η^2^_p_ = 0.03. Simple effect analysis (Figure 4) demonstrated significantly higher acceptance willingness of peers’ “driving luxury cars” in the high-effort condition (*M* = 3.05, *SD* = 0.25) versus control condition (*M* = 1.68, *SD* = 0.24), *F* (1, 176) = 16.20, *p* < 0.001, η^2^_p_ = 0.08. Conversely, acceptance of peers’ “living in hardship” showed no significant difference between control (*M* = 3.11, *SD* = 0.24) and high-effort conditions (*M* = 3.38, *SD* = 0.24), *F* (1, 176) = 0.62, *p* = 0.43. After controlling for dictator game scores as a covariate, all main effects and interactions remained significant (*ps* < 0.05).

A two-way ANOVA with relative deprivation as the dependent variable revealed a significant main effect of peers’ relative circumstances. Participants exhibited significantly greater relative deprivation under the “driving luxury cars” condition (*M* = 5.50, *SD* = 0.15) compared to the “living in hardship” condition (*M* = 2.59, *SD* = 0.15), *F* (1, 176) = 196.15, *p* < 0.001, η^2^_p_ = 0.53. Effort level also exhibited a significant main effect, with higher relative deprivation when peers expended effort (*M* = 4.49, *SD* = 0.15) versus the control condition (*M* = 3.56, *SD* = 0.14), *F* (1, 176) = 20.90, *p* < 0.001, η^2^_p_ = 0.11.

The interaction between peers’ relative circumstances and effort level was significant, *F* (1, 176) = 10.33, *p* = 0.002, η^2^_p_ = 0.06. Simple effect analysis demonstrated significantly higher relative deprivation when observing peers’ “driving luxury cars” in the high-effort condition (*M* = 6.32, *SD* = 0.20) versus the control condition (*M* = 4.66, *SD* = 0.21), *F* (1, 176) = 30.28, *p* < 0.001, η^2^_p_ = 0.15. Conversely, for peers’ “living in hardship”, relative deprivation showed no significant difference between control (*M* = 2.73, *SD* = 0.20) and high-effort conditions (*M* = 2.46, *SD* = 0.20), *F* (1, 176) = 0.92, *p* = 0.34.

A moderated mediation analysis was conducted using the PROCESS (Model 7), with peers’ relative circumstances as the independent variable, acceptance willingness as the dependent variable, relative deprivation as the mediator, and effort level as the moderator. Results revealed a significant moderating effect of effort level (Table 2), β = −1.31, *t* (176) = −3.21, *p* = 0.002, 95% *CI =* [−2.12, −0.51]. Under control conditions, the mediating effect of relative deprivation on the relationship between peers’ relative circumstances and acceptance willingness was significant, 95% *CI* = [−2.87, −1.69]. This mediating effect remained significant but diminished in the high-effort condition, 95% *CI* = [−1.89, −0.97]. Crucially, the mediating role of relative deprivation was significantly stronger in control versus high-effort conditions, 95% *CI* = [0.30, 1.44]. This indicates that individuals exhibit greater acceptance willingness of peers’ “driving luxury cars” when it results from their effort.

Experiment 4 supported Hypothesis 4, demonstrating that individuals exhibit greater acceptance willingness of peers’ “driving luxury cars” when peers exert high effort.

## 6. General Discussion

Do individuals exhibit greater reluctance toward peers’ “living in hardship” or toward peers’ “driving luxury cars”? Using lottery and bonus allocation scenarios in Experiment 1 and 2, respectively, the results demonstrated that compared to peers’ “living in hardship” (falling below the minimum requirements), participants showed lower acceptance willingness toward peers’ “driving luxury cars” (achieving the goals). This effect is attributed to peers’ “driving luxury cars” eliciting higher levels of relative deprivation in individuals. Experiment 3 and 4 manipulated situational competitiveness and peers’ effort levels, revealing that when competitiveness is low or peers expend high effort, individuals exhibit significantly greater acceptance willingness of peers’ “driving luxury cars”. Employing CMA 3.3 for single-paper meta-analysis across all four studies, we identified a robust Hedge’s g = 1.05 (95% *CI* [0.31, 1.80]) for the discrepancy in acceptance willingness toward peers’ relative circumstances. The exclusion of zero from the confidence interval confirms the effect’s statistical reliability, empirically validating our central thesis: individuals exhibit significantly lower acceptance of peers’ “driving luxury cars” compared to peers’ “living in hardship”.

### 6.1. Research Implications

First, this research integrates the Tri-Reference Point theory with social comparison theory, exploring individuals’ acceptance willingness toward others’ relative standing within a social comparison context. As established by the Tri-Reference Point theory, people generally assign greater psychological weight to the minimum requirement than the goal (e.g., [20]; [38]; [45]). Regarding personal reference points, for instance, individuals perceive salaries falling below the minimum requirement as particularly unacceptable compared to attaining a goal, resulting in lower fairness and satisfaction evaluations ([50]). To avoid falling below the minimum requirement, individuals exhibit heightened risk-seeking tendencies and stronger turnover intentions ([38]; [45]). Concerning others’ reference points, individuals show greater willingness to help others avoid falling below the minimum requirement than to assist them in achieving a goal ([20]). Based on this theoretical framework, people should exhibit greater aversion to peers falling below the minimum requirement (“living in hardship”) than to peers achieving goals (“driving luxury cars”). However, in contexts such as judging whether it is less desirable to see a peer ‘living in hardship’ or ‘driving a luxury car,’ individuals do not focus solely on the peer’s absolute situation, that is, they do not merely consider whether the peer has succeeded or failed. Instead, within the framework of social comparison, they consider the disparity between themselves and the peer, focusing fundamentally on their relative circumstances.

This research reveals a reversal in the psychological weighting of goals versus minimum requirements within the Tri-Reference Point theory under social comparison. Compared to peers’ “living in hardship” (falling below the minimum requirements), individuals exhibit significantly lower acceptance willingness toward peers “driving luxury cars” (achieving goals). This indicates that in social comparison contexts, people prioritize others’ goal reference points over their minimum requirement reference points. Theoretically, this finding constitutes a significant extension of Tri-Reference Point theory.

Second, this research extends the literature on changes in others’ circumstances. Unlike prior studies primarily examining unidirectional shifts in circumstance (e.g., [20]; [38]) or redistribution scenarios involving “improvement for low-status individuals and reduction for high-status individuals” to narrow societal disparities ([3]), this study investigates circumstances shifting in opposing directions: “high-performing peers get richer, while those already struggling sink deeper into hardship”. Crucially, we found that under social comparison, individuals express greater concern about peers achieving sudden success than about peers falling into adversity. This diverges from [7]’s ([7]) conclusion, which demonstrated that when downward-comparison targets worsen and upward-comparison targets improve, individuals experience sympathy and envy, respectively, with sympathy surpassing envy in intensity—suggesting greater aversion to others’ misfortune (“living in hardship”). We posit this discrepancy stems from our research’s closer proximity to real-world context. Crucially, [7] ([7]) also observed that when comparisons were highly relevant, envy intensified more markedly than sympathy. Their laboratory-based fictional scenarios provided weak motivational triggers for social comparison, whereas our ecologically valid lottery and bonus contexts prompted stronger comparative motivations and greater likelihood of envy toward peers’ goal achievement. Future research could directly compare the intensity of sympathy versus envy toward peers below or above minimum requirements and goals.

Moreover, this research enriches the theoretical framework concerning motivations underlying social comparison. The motivational perspective of social comparison posits that the intensity of comparisons influences individuals’ motivational states. As downward comparison intensifies, individuals engage in self-enhancement, perceiving themselves as sufficiently competent and consequently reducing effort expenditure (coasting). During upward comparison, however, individuals develop self-improvement motivation, striving to achieve goals through increased effort (pushing). Notably, when upward comparison reaches extreme levels, individuals perceive an insurmountable gap with comparators, leading to complete disengagement from effort ([13]). Our findings reveal that the inflection point where motivational intensity shifts may relate to the goal reference point. The Tri-Reference Point theory contends that crossing reference points triggers qualitative shifts in individuals’ circumstances ([38]). Individuals construct minimum requirements and goals based on their status quo positioning. When others achieve their goals, the qualitative disparity between these successful peers and individuals near the status quo becomes psychologically salient. This perception makes such disparities difficult to accept and undermines motivation for catch-up efforts. Consequently, when comparators achieve individuals’ goals—exemplified by “driving luxury cars”—motivation to exert effort begins to decline.

Finally, our research provides empirical substantiation for the significance of effort in the workplace, such as working diligently and conscientiously completing tasks assigned by supervisors. As established by prior studies, individuals assign greater value and meaning to tasks requiring substantial effort ([24]). Crucially, Experiment 4 revealed that when others achieve success through their efforts, individuals are relatively more likely to accept such a situation. This has enriched the literature on effort, revealing that people can recognize both the value of their own efforts, and the value of others’ efforts. In addition, it should be noted that our investigation does not encompass all types of effort. Future research could examine other forms of exertion, such as cognitive effort, to explore whether individuals are more accepting of peers’ “driving luxury cars” when these are perceived as resulting from intense cognitive labor.

This research also offers significant practical implications. On the one hand, our findings reveal that individuals exhibit lower acceptance willingness of peers’ “driving luxury cars” compared to “living in hardship”. This suggests people do not always welcome peers’ upward mobility. Humans are “prisoners of comparison”—social comparisons inevitably emerge wherever people interact ([27]). Our work illuminates the psychological conflicts individuals experience when confronted with changes in others’ relative standing during social comparisons. More significantly, low acceptance of peers’ success (“driving luxury cars”) may trigger detrimental competition, thereby impairing overall team performance. For team administrators and policymakers, recognizing how shifts in peers’ relative circumstances affect psychological responses can inform the design of interventions that emphasize effort and fairness while dismantling environments of relentless competition, ultimately advancing social harmony ([22]). Importantly, this study demonstrates that highlighting the effort behind success and reducing situational competitiveness can mitigate relative deprivation, thereby increasing acceptance of peers’ success.

### 6.2. Limitations and Direction for Future Research

First, this study treats peers’ “living in hardship” as falling below the minimum requirements and peers’ “driving luxury cars” as achieving goals within the Tri-Reference Point theory. However, on the one hand, the correspondence between “living in hardship” and falling below the minimum requirements, and “driving luxury cars” and achieving goals, may not be strictly bidirectional. On the other hand, the scenario descriptions in this study did not explicitly use the concepts of minimum requirement or goal. Future research could more directly and explicitly manipulate falling below the minimum requirements and achieving the goals.

Second, this study exclusively examined individuals’ acceptance willingness toward peers’ “living in hardship” and “driving luxury cars”. Within the Tri-Reference Point theory framework, this corresponds to investigating acceptance of changes near peers’ minimum requirement and goal reference points, but neglects attitudes toward shifts around the status quo. The perspective of rank reversal aversion posits that people prefer maintaining stable relative positions to preserve group equality, expressing aversion to status reversals ([17]; [44]). Consequently, when peers’ situations directly surpass or fall below one’s own status quo position—thereby triggering rank reversal—individuals may exhibit greater reluctance toward such changes. This suggests that under social comparison, the psychological weighting of the three reference points would prioritize status quo > goal > minimum requirement. Future research should incorporate peers’ situational shifts around the status quo reference point, enabling a more systematic exploration of psychological weight adjustments across all three reference points in social comparison contexts.

Third, while our research frames reluctance toward peers’ “driving luxury cars” as potentially detrimental to team harmony, this phenomenon may not invariably yield negative consequences. Envy—particularly benign envy—can functionally motivate self-improvement efforts ([36]). Therefore, applications of these findings require context-specific implementation: maintaining collective relationships while strategically leveraging comparative dynamics to stimulate individual growth motivation. In addition, the phenomena of “living in hardship” and “driving luxury cars” extend beyond economic domains to comparative contexts in academic performance, interpersonal relationships and health outcomes. Individuals’ minimum requirements and goals demonstrate significant heterogeneity across social strata—for instance, a homeless individual might perceive a nutritious meal as goal attainment, whereas an academically elite student could consider median rankings as falling below minimum requirements. Future research should investigate attitudes toward peers’ relative positions across non-economic domains, particularly examining how status continually shape perceptions of comparative disadvantage and advantage.

Finally, the participants recruited for this study all came from China, a typical collectivistic culture. Therefore, the conclusions drawn here might differ in individualistic cultural contexts. For instance, prior research suggests that individuals in individualistic cultures engage in less social comparison ([4]; [10]), which could lead to greater acceptance of peers’ success. Future research should directly investigate these findings across diverse cultural settings to explore the potential cultural boundaries of the effects observed in the present study. Furthermore, there are still other potential explanations for the research. For instance, people strive to maintain a positive self-concept ([2]; [40]). Individuals might fear being perceived as incompetent, evoking feelings of personal threat ([31]), which in turn could lead to reluctance in accepting peers’ “driving luxury cars”. Future research could productively examine these and other potential alternative explanations.

## 7. Conclusions

(1) Individuals exhibit significantly lower acceptance willingness toward peers’ “driving luxury cars” (achieving the goals) compared to peers’ “living in hardship” (falling below the minimum requirements). (2) This phenomenon arises primarily because peers’ “driving luxury cars” triggers higher levels of relative deprivation than peers’ “living in hardship”. (3) When situational competitiveness is low or peers expend high effort, relative deprivation decreases, consequently increasing acceptance willingness toward peers’ “driving luxury cars”.

## Figures and Tables

**Figure 1 behavsci-15-01395-f001:**
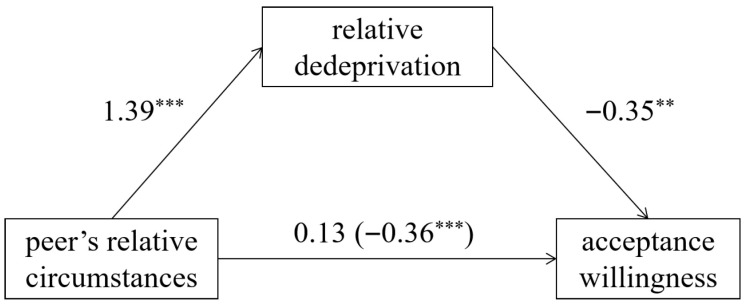
The mediating role of relative deprivation in Experiment 1. (Note: Values represent standardized regression coefficients; ** *p* < 0.01, *** *p* < 0.001).

**Figure 2 behavsci-15-01395-f002:**
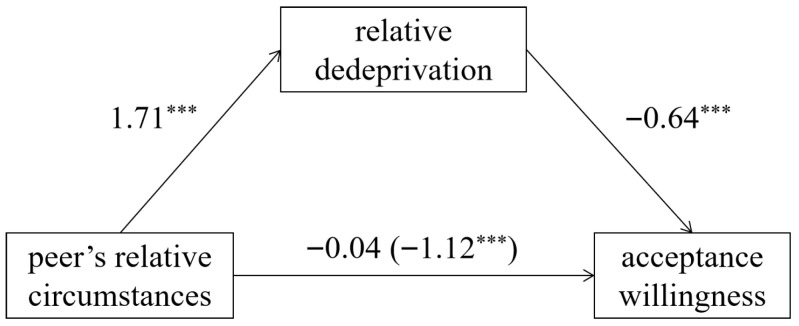
The mediating role of relative deprivation in Experiment 2. (Note: Values represent standardized regression coefficients; *** *p* < 0.001).

**Figure 3 behavsci-15-01395-f003:**
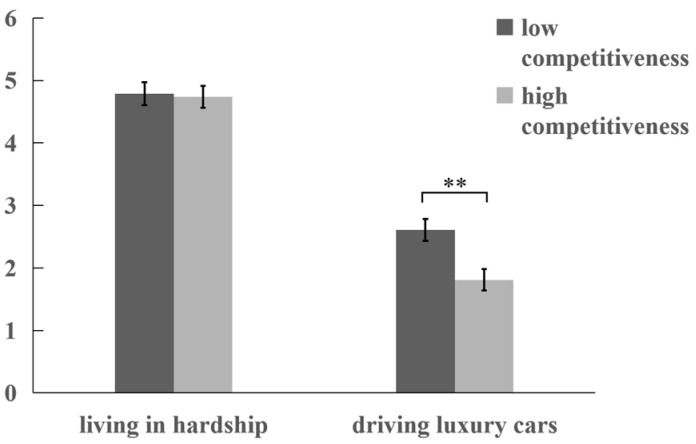
Participants’ acceptance willingness across experimental conditions in Experiment 3. (Note. Error bars represent standard errors; ** *p* < 0.01).

**Figure 4 behavsci-15-01395-f004:**
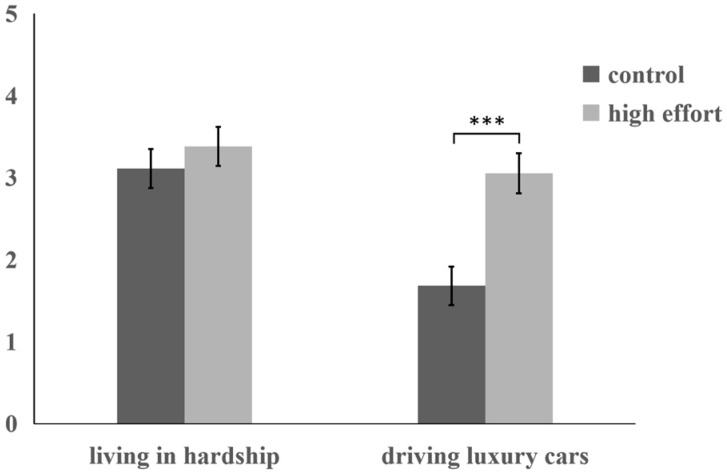
Participants’ acceptance willingness across experimental conditions in Experiment 4. (Note. Error bars represent standard errors; *** *p* < 0.001).

**Table 1 behavsci-15-01395-t001:** Results of moderated mediation analysis in Experiment 3.

Competitiveness	Effect	*SE*	95% *CI*
low	−1.38	0.21	[−1.77, −0.99]
high	−2.41	0.27	[−2.91, −1.85]

**Table 2 behavsci-15-01395-t002:** Results of moderated mediation analysis in Experiment 4.

Effort Level	Effect	*SE*	95% *CI*
control	−2.27	0.30	[−2.87, −1.69]
high	−1.42	0.23	[−1.89, −0.97]

## Data Availability

Data supporting reported results are available from the authors on request.

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
