# Peer review of "Are People More Averse of Their Peers Living in Hardship or Driving Luxury Cars? Individuals’ Willingness to Accept Their Peers’ Relative Circumstances"

_behavsci, 2025, doi:10.3390/bs15101395_

Round 1

Reviewer 1 Report

Comments and Suggestions for Authors

I have carefully reviewed the manuscript, titled “Are People More Averse of Their Peers' Living in Hardship or Driving Luxury Cars? Individuals' Willingness to Accept Their Peers' Relative Circumstances”. It consisted of four experiments, which are really interesting and valuable.

The study has been revised and some strong aspects. Yet, I would like to ask the authors to address some points in order to improve their paper.

Introduction:

1) Can you describe more thoroughly the psychological feelings related to "living in hardship" or "driving luxury cars? What are its underlying mechanisms (p. 1)

2) Please, compare the cultural context of fearing peers "living in hardship" or "driving 59 luxury cars", (p. 2). How does culture diversify these two approaches?

3) P. 3 – what theory could underlie the moderating effects of situational competitiveness and peers' effort level?

4) The result are properly presented. Congratulations!

5) The sentence on p. 10 is rather fuzzy: “However, in actuality, when evaluating their aversion toward peers' "living in hardship" versus "driving luxury cars", individuals do not solely focus on others' absolute situations.” Absolute situations? What does it mean?

6) Can you elaborate on the following statement: “Finally, our research provides empirical substantiation for the significance of effort.” (p. 11). What kind of effort do you have in mind? Cognitive activities? Social actions?

7) P. 11 – I would also add some interpretation and explanation regarding why people do not value their own effort (lines 460-461). It sound rather counterintuitive as people rather tend to appreciate their endeavours.

Author Response

Responses to the Reviewer 1s comments

I have carefully reviewed the manuscript, titled “Are People More Averse of Their Peers' Living in Hardship or Driving Luxury Cars? Individuals' Willingness to Accept Their Peers' Relative Circumstances”. It consisted of four experiments, which are really interesting and valuable. The study has been revised and some strong aspects. Yet, I would like to ask the authors to address some points in order to improve their paper.

Responses: We are deeply grateful for your time and effort in reviewing our manuscript and for providing valuable positive feedback and constructive suggestions. Your insightful and thorough review has been instrumental in helping us improve the quality of this paper. We have carefully revised the manuscript and provided additional clarifications based on each of your comments. Below, we provide a point-by-point response to the issues you raised. The revisions we made in the manuscript were marked in blue font. We hope our revisions and responses will meet with your approval. 

1) Can you describe more thoroughly the psychological feelings related to "living in hardship" or "driving luxury cars? What are its underlying mechanisms (p. 1)

Responses: We sincerely thank you for this excellent suggestion.

Upon reviewing the introduction, we recognized that the psychological meanings of "living in hardship" and "driving luxury cars" were initially outlined too briefly, which might hinder readers' understanding. Besides, it is important to clarify that, in this study, we examine these concepts from a third-person observer's perspective rather than a first-person one. In other words, our focus is on the psychological reactions triggered by others (not oneself) who are either living in hardship or driving luxury cars.

To address this, we have added a more detailed description of the psychological connotations of these phrases in the introduction. The added text reads: "The Chinese saying, 'We're not only afraid of our peers living in hardship, but also afraid of them driving luxury cars,' reveals complex attitudes toward others' situations. On one hand, witnessing peers suffer can evoke sympathy, driven by concern for their well-being. On the other hand, observing peers achieve success while one remains in a mediocre state can trigger feelings of envy (Batson et al., 1981; Festinger, 1954)."

Batson, C. D., Duncan, B. D., Ackerman, P., Buckley, T., & Birch, K. (1981). Is empathic emotion a source of altruistic motivation? Journal of Personality and Social Psychology, 40(2), 290–302.

Festinger, L. (1954). A theory of social comparison processes. Human relations, 7(2), 117–140.

2) Please, compare the cultural context of fearing peers "living in hardship" or "driving  luxury cars", (p. 2). How does culture diversify these two approaches?

Responses: Thank you for this insightful suggestion.

You are absolutely correct that the level of acceptance towards peers "living in hardship" or "driving luxury cars" may vary across cultures. We have addressed this point in the Discussion section by adding a more detailed explanation and encouraging future cross-cultural research to examine potential cultural differences in our findings. The specific addition is as follows:

"The participants recruited for this study all came from China, a typical collectivistic culture. Therefore, the conclusions drawn here might differ in individualistic cultural contexts. For instance, prior research suggests that individuals in individualistic cultures engage in less social comparison (Baldwin & Mussweiler, 2018; Cheng et al., 2021), which could lead to greater acceptance of peers' success. Future research should directly investigate these findings across diverse cultural settings to explore the potential cultural boundaries of the effects observed in the present study."

Baldwin, M., & Mussweiler, T. (2018). The culture of social comparison. Proceedings of the National Academy of Sciences, 115(39), E9067–E9074.

Cheng, J., Burke, M., & de Gant, B. (2021). Country differences in social comparison on social media. Proceedings of the ACM on Human-Computer Interaction, 4(CSCW3), 1–26.

3) P. 3 – what theory could underlie the moderating effects of situational competitiveness and peers' effort level?

Responses: Thank you for this expert suggestion.

Consistent with the derivation of our hypotheses for the main and mediating effects, our hypotheses regarding the moderating effects were also grounded in the theoretical framework of social comparison. We acknowledge that in the previous version of the manuscript, our explanation may have been unclear and did not sufficiently highlight the role of this framework. In reality, both the competitiveness of the situation and the level of a peer's effort are key factors influencing social comparisons, and they consequently affect individuals' willingness to accept peers who are either "living in hardship" or "driving luxury cars." We have revised the sections detailing the derivation of the two moderating effect hypotheses to better emphasize their theoretical foundation. The modified content is as follows:

For Hypothesis 3: "The competitiveness between comparison targets is a salient feature of social comparison (Garcia et al., 2013). In highly competitive situations, individuals exhibit a stronger tendency to compare themselves with others (Stapel & Koomen, 2005). Conversely, When the situational competitiveness is low, individuals exhibit reduced motivation to compare themselves with others (Boecker et al., 2022), and are less likely to engage in harmful behaviors driven by envy (Behler et al., 2020). Research also indicates that competitiveness can lead people to perceive others' successes as being achieved at their own expense, thereby triggering a stronger sense of relative deprivation (Ongis & Davidai, 2022; To et al., 2020). Therefore, under the less competitive situations, the relative deprivation elicited by a peer's success is weaker, leading to greater acceptance of peers who are 'driving luxury cars'."

For Hypothesis 4: "During social comparisons, individuals consider not only the comparison of outcomes but also the comparison of inputs between themselves and others (Adams, 1963). People attribute moral value to effort, perceiving those who work hard as having superior moral character (Celniker et al., 2023), and are thus more accepting of success and resulting inequality when it is perceived as earned through effort (Cui & He, 2022; Ericsson et al., 1993). Accordingly, we hypothesize that if a peer's luxury car is perceived as earned through effort, the ensuing sense of relative deprivation will be lower, leading to greater acceptance of the peer 'driving a luxury car'."

Adams, J. S. (1963). Towards an understanding of inequity. The journal of abnormal and social psychology, 67(5), 422–436.

Behler, A. M. C., Wall, C. S., Bos, A., & Green, J. D. (2020). To help or to harm? Assessing the impact of envy on prosocial and antisocial behaviors. Personality and Social Psychology Bulletin, 46(7), 1156–1168.

Boecker, L., Loschelder, D. D., & Topolinski, S. (2022). How individuals react emotionally to others’(mis) fortunes: A social comparison framework. Journal of Personality and Social Psychology, 123(1), 55–83.

Celniker, J. B., Gregory, A., Koo, H. J., Piff, P. K., Ditto, P. H., & Shariff, A. F. (2023). The moralization of effort. Journal of Experimental Psychology: General, 152(1), 60–79.

Cui, W., & He, Y. (2022). Cognition of opportunity inequality and residents' happiness: An empirical study on family background, luck and personal effort. Journal of Guizhou University of Finance and Economics, 40(2), 79–88.

Ericsson, K. A., Krampe, R. T., & Tesch-Römer, C. (1993). The role of deliberate practice in the acquisition of expert performance. Psychological review, 100(3), 363–406.

Garcia, S. M., Tor, A., & Schiff, T. M. (2013). The psychology of competition: A social comparison perspective. Perspectives on psychological science, 8(6), 634–650.

Ongis, M., & Davidai, S. (2022). Personal relative deprivation and the belief that economic success is zero-sum. Journal of Experimental Psychology: General, 151(7), 1666–1680.

Stapel, D. A., & Koomen, W. (2005). Competition, cooperation, and the effects of others on me. Journal of personality and social psychology, 88(6), 1029–1038.

To, C., Kilduff, G. J., & Rosikiewicz, B. L. (2020). When interpersonal competition helps and when it harms: An integration via challenge and threat. Academy of Management Annals, 14(2), 908–934.

4) The result are properly presented. Congratulations!

Responses: Thank you for your positive feedback on our research!

  • The sentence on p. 10 is rather fuzzy: “However, in actuality, when evaluating their aversion toward peers' "living in hardship" versus "driving luxury cars", individuals do not solely focus on others' absolute situations.” Absolute situations? What does it mean?

Responses: Thank you for this valuable suggestion!

In our manuscript, "absolute situations" refer to an individual's objective circumstances without invoking social comparison, which stands in contrast to "relative situations" that arise specifically from comparative evaluations. To illustrate, considering a person's "absolute situation" involves examining their objective wealth, for instance, that they possess 1000. In contrast, considering their "relative situation" involves evaluating the comparative difference between their wealth and one's own, for example, that they have 1000 while I have ¥500. Our findings demonstrate that within the context of social comparison, individuals are more concerned with their relative standing compared to others rather than merely focusing on the others' absolute success or failure.

Your comment also made us realize that our original description of this finding was not sufficiently clear. Therefore, we have refined the explanation of this result as follows: "However, in contexts such as judging whether it is less desirable to see a peer 'living in hardship' or 'driving a luxury car,' individuals do not focus solely on the peer's absolute situation, that is, they do not merely consider whether the peer has succeeded or failed. Instead, within the framework of social comparison, they consider the disparity between themselves and the peer, focusing fundamentally on their relative circumstances."

6) Can you elaborate on the following statement: “Finally, our research provides empirical substantiation for the significance of effort.” (p. 11). What kind of effort do you have in mind? Cognitive activities? Social actions?

Responses: Thank you for this valuable suggestion.

In this study, the term "effort" specifically refers to the exertion individuals invest in their work to achieve higher performance, manifested as sustained dedication to job-related and career advancement activities. Our findings demonstrate that individuals not only value the worth of their own efforts but also recognize the value of efforts exerted by others. Accordingly, we have clarified the specific type of effort examined in this research as follows: "Finally, our research provides empirical substantiation for the significance of effort in the workplace, such as working diligently and conscientiously completing tasks assigned by supervisors."

At the same time, we acknowledge that the conceptualization of effort in our experimental manipulations is not exhaustive. Beyond behavioral effort in social actions, individuals often expend cognitive effort, such as striving to devise solutions to problems. We have therefore added the following clarification in the Discussion section: "Besides, it should be noted that our investigation does not encompass all types of effort. Future research could examine other forms of exertion, such as cognitive effort, to explore whether individuals are more accepting of peers' 'driving luxury cars' when these are perceived as resulting from intense cognitive labor."

  • 11 – I would also add some interpretation and explanation regarding why people do not value their own effort (lines 460-461). It sound rather counterintuitive as people rather tend to appreciate their endeavours.

Responses: We sincerely appreciate your suggestion!

In Study 4, we manipulated the level of effort exerted by peers to examine its influence on participants' acceptance of peers who drive luxury cars. The results indicated that when peers were perceived as having achieved success through hard work, participants showed greater acceptance of their "driving luxury cars". This suggests that individuals are able to recognize the important role of effort in their peers' success.

We wish to clarify that our original statement—"Experiment 4 demonstrates that people not only value their own effort, but also acknowledge the worth of peers' effort behind their success"—may have been misleading. It was not our intention to imply that people do not value the outcomes of their own efforts. Rather, we aimed to emphasize that people can appreciate both their own and others' efforts. To prevent similar misunderstandings, we have revised the relevant passage as follows: "Experiment 4 revealed that when others achieve success through effort, individuals are more likely to accept such outcomes. This has enriched the literature on effort, revealing that people can recognize both the value of their own efforts, and the value of others' efforts."

Reviewer 2 Report

Comments and Suggestions for Authors

The article examines the effects of upward and downward social comparison in relation to reference point theory. Participants are more reluctant to accept situations where a peer achieves greater gains with better rewards than they do, compared to a condition where others perform worse. The feeling of relative deprivation plays a mediating role. A competitive situation makes a peer's success less acceptable, while the peer's effort to achieve their goals reduces this reluctance to accept others' success.

These 4 studies are well conducted. However, I would like to make some comments: first of all, from a statistical point of view, it seems to me that the authors made a mistake with the g*power analysis; in particular, the chosen effect size 0.5 is probably 0.25. For all the studies, I do not find the same recommendations for sample sizes when introducing the given criteria. This needs to be verified. Then, for different studies (3 and 4), manipulation control covariates are used (dictator game measurement; it should be more detailed and how measure is calculated, like the ZSB competitiveness item; which item is used?), but no information is given on their significance and the regression coefficients obtained, as well as the presence or absence of interaction effects with the covariates. Regarding these covariates, no comment is made on the role of altruism and conformity in relation to these control measures and the reasons why the results might be affected.

The feeling of relative deprivation (RP) is an interesting mediating variable that the authors explore. It should also be noted that the feeling of RP is related to the fact that people feel they deserve the object they did not obtain and that they find this unfair (i.e. they are competent and deserve this gain). Another element that could be raised in the general discussion is that the feeling of personal threat (proof of one's incompetence, ineffectiveness etc.) could also play a mediating role in the fact that upward social comparison is more threatening to the self than downward comparison. This element is not incompatible with the role of RP, and this could make sense when people do not accept being considered as not competent or not efficient.

Author Response

Responses to the Reviewer 2s comments

The article examines the effects of upward and downward social comparison in relation to reference point theory. Participants are more reluctant to accept situations where a peer achieves greater gains with better rewards than they do, compared to a condition where others perform worse. The feeling of relative deprivation plays a mediating role. A competitive situation makes a peer's success less acceptable, while the peer's effort to achieve their goals reduces this reluctance to accept others' success.

These 4 studies are well conducted. However, I would like to make some comments:

Responses: We sincerely appreciate your positive feedback on our manuscript. Your comments have also helped us recognize several areas in the manuscript that required further improvement. We have carefully revised the paper based on your valuable insights, and our detailed responses are provided below. The revisions we made in the manuscript were marked in blue font. Thank you once again for your thoughtful and constructive suggestions.

1. First of all, from a statistical point of view, it seems to me that the authors made a mistake with the g*power analysis; in particular, the chosen effect size 0.5 is probably 0.25. For all the studies, I do not find the same recommendations for sample sizes when introducing the given criteria. This needs to be verified.

Responses: We sincerely appreciate your meticulous suggestion.

You are absolutely correct. We made an error in the manuscript by incorrectly reporting the effect size for the ANOVA as f= 0.5, when it should have been f= 0.25. We have corrected this throughout the manuscript. For your reference, we used a consistent standard for effect sizes in the a priori power analyses for all four studies: a medium effect size of Cohen's d= 0.5 for independent-samples t-tests and a medium effect size of f= 0.25 for the two-way ANOVA. Thank you once again for your keen attention to detail, which has been invaluable in helping us correct this oversight.

2. Then, for different studies (3 and 4), manipulation control covariates are used (dictator game measurement; it should be more detailed and how measure is calculated, like the ZSB competitiveness item; which item is used?), but no information is given on their significance and the regression coefficients obtained, as well as the presence or absence of interaction effects with the covariates. Regarding these covariates, no comment is made on the role of altruism and conformity in relation to these control measures and the reasons why the results might be affected.

Responses: Thank you for this highly professional suggestion.

To rule out potential confounding effects from extraneous variables, we measured additional factors in Experiments 3 and 4 that could potentially influence the results.

The altruism of participants might affect the study's outcomes. For instance, individuals with high levels of altruism are generally more willing to help others in distress and are also more accepting of others' successes (Batson et al., 1981). It was therefore necessary to control for participants' innate altruistic tendencies to isolate this variable's influence. The dictator game is a widely used paradigm for measuring individual altruism (Forsythe et al., 1994). Consequently, in both Experiments 3 and 4, after completing the main experimental task, participants engaged in a dictator game exercise. Specifically, they were asked to imagine completing a task with a stranger, to which both had contributed equal effort and time. A total reward of ¥10 was to be divided between the participant and the stranger, and the participant decided how much to allocate to the other person. We have supplemented the Method sections with the following details:

​​For Experiment 3: "Furthermore, since participants' inherent altruism could influence the results. For example, highly altruistic individuals are more willing to help those in hardship and more accepting of others' success(Batson et al., 1981). We also measured altruistic tendencies using a dictator game task (Forsythe et al., 1994). Specifically, participants imagined completing a task with a stranger with equal effort and time invested. They were informed that a total reward of ¥10 was to be divided between themselves and the stranger, and they indicated how much they would allocate to the other party."

​​For Experiment 4: "Finally, participants completed the same measures of relative deprivation and situational acceptance willingness as in previous studies, along with the same altruism measurement (i.e., the dictator game) used in Experiment 3."

It is important to note that in Experiment 3, we also measured zero-sum beliefs to validate the effectiveness of the competitiveness manipulation. Prior research indicates that situational competitiveness elicits zero-sum beliefs (Ongis & Davidai, 2022). Accordingly, we assessed zero-sum beliefs using the item, "When some people are getting poorer, it means that other people are getting richer." (Różycka-Tran et al., 2015), rated on a 1-7 scale. This content has been added to the experimental design section of Experiment 3 as follows:

"Consistent with prior findings that situational competitiveness promotes zero-sum beliefs (Ongis & Davidai, 2022), we included a zero-sum belief item ('When some people are getting poorer, it means that other people are getting richer.') on a 1-7 point scale (Różycka-Tran et al., 2015) to serve as a manipulation check."

Furthermore, due to social desirability concerns, participants might have deliberately responded in a way that indicated they wished for their peers' success rather than hardship. Therefore, in Study 1, we also measured participants' social desirability as a covariate. After completing all tasks, participants filled out a social desirability scale (Wei et al., 2015). We have added the relevant details to the Method section of Study 1: "Additionally, social desirability bias could influence participants' choices, leading them to appear more averse to peers living in hardship and more accepting of peers driving luxury cars. Consequently, after finishing all tasks, participants completed a short-form social desirability scale to be included as a covariate (Wei et al., 2015)."

Batson, C. D., Duncan, B. D., Ackerman, P., Buckley, T., & Birch, K. (1981). Is empathic emotion a source of altruistic motivation? Journal of Personality and Social Psychology, 40(2), 290–302.

Forsythe, R., Horowitz, J. L., Savin, N. E., & Sefton, M. (1994). Fairness in simple bargaining experiments. Games and Economic behavior, 6(3), 347–369.

Ongis, M., & Davidai, S. (2022). Personal relative deprivation and the belief that economic success is zero-sum. Journal of Experimental Psychology: General, 151(7), 1666–1680.

Różycka-Tran, J., Boski, P., & Wojciszke, B. (2015). Belief in a zero-sum game as a social axiom: A 37-nation study. Journal of Cross-Cultural Psychology, 46(4), 525–548.

Wei, J., Han, H. F., Zhang, C. Y., Sun, L. J., & Zhang, J. F. (2015). Reliability and Validity of the Marlowe-crowne Social Desirability Scale in Middle School Students. Chinese Journal of Clinical Psychology, 23(4), 585–589.

3. The feeling of relative deprivation (RP) is an interesting mediating variable that the authors explore. It should also be noted that the feeling of RP is related to the fact that people feel they deserve the object they did not obtain and that they find this unfair (i.e. they are competent and deserve this gain).

Responses: Thank you for this highly valuable suggestion.

We recognize that, as you rightly pointed out, the introduction did not sufficiently elaborate on the meaning of relative deprivation, merely mentioning the circumstances under which it arises. A clear understanding of this concept is indeed crucial for clarifying the proposed mediating mechanism. Therefore, we have incorporated the definition you referenced into the manuscript as follows:

"The feeling of relative deprivation is related to the fact that people feel they deserve the object they did not obtain and that they find this unfair (i.e., they are competent and deserve this gain)." For instance, when a peer achieves success, an individual may feel discontent regarding their own lack of comparable success, thereby experiencing relative deprivation."

4. Another element that could be raised in the general discussion is that the feeling of personal threat (proof of one's incompetence, ineffectiveness etc.) could also play a mediating role in the fact that upward social comparison is more threatening to the self than downward comparison. This element is not incompatible with the role of RP, and this could make sense when people do not accept being considered as not competent or not efficient

Responses: Thank you for this insightful suggestion.

You are absolutely correct that the "feeling of personal threat" constitutes a plausible alternative explanation. Indeed, while our study specifically examined relative deprivation as the mediating mechanism, we cannot rule out the possibility of other explanatory pathways underlying the observed effect of being more averse to peers "driving luxury cars". Social psychological research indicates that individuals are motivated to maintain a positive self-concept (Alicke & Sedikides, 2009; Wang & Shen, 2022). Consequently, when an individual's situation is markedly worse than a peer's, it may be perceived as an indicator of low competence, thereby threatening the self-concept (Park et al., 2023). Therefore, feelings of personal threat could indeed serve as an alternative account. That is, individuals might be more reluctant to see a peer driving a luxury car due to concerns about the personal threat it poses.We have added a discussion regarding personal threat in the relevant section, as follows:

"Furthermore, other potential explanations exist. For instance, people strive to maintain a positive self-concept (Alicke & Sedikides, 2009; Wang & Shen, 2022). Individuals might fear being perceived as incompetent, evoking feelings of personal threat (Park et al., 2023), which in turn could lead to reluctance in accepting peers' 'driving luxury cars'. Future research could productively examine these and other potential alternative explanations."

Wang, X. X., & Shen, Y. F. (2022). How to cope with the threat to moral self? The perspectives of memory bias in moral contexts. Advances in Psychological Science, 30(7), 1604–1611.​

Alicke, M. D., & Sedikides, C. (2009). Self-enhancement and self-protection: What they are and what they do. European review of social psychology, 20(1), 1–48.

Park, L. E., Naidu, E., Lemay, E. P., Canning, E. A., Ward, D. E., Panlilio, Z., & Vessels, V. (2023). Social evaluative threat across individual, relational, and collective selves. In Advances in experimental social psychology, 68, 139–222.